# Analysis of Prognostic Factors for Internal Carotid Artery Invasion by Nasopharyngeal Carcinoma

**DOI:** 10.3390/cancers17030488

**Published:** 2025-02-01

**Authors:** Ching-Feng Lien, Shyh-An Yeh, Chiu-Shih Cheng, Feng-Yu Chiang, Tzer-Zen Hwang, Bi-He Cai, Chih-Yi Liu, Hsu-Huei Weng, Chia-Chi Chen, Meng-Che Hsieh

**Affiliations:** 1Department of Otolaryngology-Head and Neck Surgery, E-Da Hospital, I-Shou University, Kaohsiung 82445, Taiwan; 2School of Medicine, College of Medicine, I-Shou University, Kaohsiung 82445, Taiwanbigbiha@isu.edu.tw (B.-H.C.); 3Department of Radiation Oncology, E-Da Hospital, I-Shou University, Kaohsiung 82445, Taiwan; 4Department of Neuroradiology, E-Da Hospital, I-Shou University, Kaohsiung 82445, Taiwan; 5School of Medicine for International Students, College of Medicine, I-Shou University, Kaohsiung 82445, Taiwan; 6Nursing Department, E-Da Hospital, I-Shou University, Kaohsiung 82445, Taiwan; 7Department of Diagnostic Radiology, Chang Gung Memorial Hospital, Chiayi, Chang Gung University College of Medicine, Taoyuan City 33302, Taiwan; 8Department of Pathology, E-Da Hospital, I-Shou University, Kaohsiung 82445, Taiwan; 9Department of Hematology-Oncology, E-Da Cancer Hospital, Kaohsiung 82445, Taiwan

**Keywords:** nasopharyngeal cancer, chemoradiotherapy, internal carotid artery invasion, prognostic factor, survival, prognosis

## Abstract

Our study aimed to analyze prognostic factors for internal carotid artery (ICA) invasion by nasopharyngeal cancer. Our results demonstrated that the ICA invasion group showed a worse prognosis compared to the no ICA invasion group. Multivariate analysis identified that the neutrophil–lymphocyte ratio (NLR) was an independently prognostic factor for overall survival and for disease-specific survival. Hence, we concluded that locally advanced NPC patients with ICA invasion have a miserable outcome and NLR represents a significant prognostic factor that impacts treatment decisions and survival.

## 1. Introduction

Nasopharyngeal carcinoma (NPC) is the most common malignancy in the nasopharynx. Its incidence varies significantly across different geographic regions and populations. The highest rates of NPC are found in Southeast Asia, particularly in southern China, Malaysia, and Singapore [1]. In these areas, the age-standardized incidence can reach as high as 20 to 50 cases per 100,000 individuals, and 5 cases per 100,000 individuals in Western countries [2]. The incidences in men are roughly three times higher than those in women [3]. Meanwhile, the incidences of keratinizing NPC and non-keratinizing NPC were different between Southeast Asia and Western countries. In endemic areas, 25% of patients were keratinizing NPC and 75% were non-keratinizing NPC [4], while in non-endemic areas, 45% were differentiated NPC (including keratinizing and non-keratinizing) and 55% were undifferentiated NPC [5]. Epidemiological trends in the past decade have shown that its incidence has declined gradually but progressively, and mortality has been reduced substantially [6]. For patients with locally advanced NPC, combining radiation with chemotherapy (CCRT) significantly improves treatment outcomes [7]. Furthermore, induction chemotherapy followed by CCRT has become the standard of care in many regions, enhancing local control and reducing the likelihood of distant failures [8]. Moreover, the widespread application of intensity-modulated radiotherapy and optimization of chemotherapy strategies (induction, concurrent, adjuvant) have contributed to improved survival with reduced toxicities [9]. In general, the prognosis for NPC can be favorable, with 5-year survival rates reaching more than 90% in patients with stage I disease and roughly 50% in patients with stage IV disease [10].

The management of NPC with internal carotid artery (ICA) invasion is very challenging [11]. The previous literature demonstrated that survival was significantly compromised in NPC patients with ICA invasion across all tumor stages [12]. The major cause of death in NPC patients with ICA invasion was a massive hemorrhage after CCRT, resulting in shortened survival [13]. In patients whose primary tumor was pharyngeal recess with carotid artery invasion, ≥270° led to a higher risk of massive hemorrhage after radiotherapy [14]. Hence, prophylactic ICA management seems important for NPC patients with ICA invasion. Liu et al. showed that bypass grafting in recurrent nasopharyngeal carcinoma patients with internal carotid artery invasion is an effective treatment for recurrent NPC patients with internal carotid artery invasion [15]. Chen et al. presented that salvage endoscopic nasopharyngectomy combined with ICA pretreatment allows the feasible and effective resection of NPC lesions adjacent to the ICA [16]. Although treatment advances for NPC patients with ICA invasion, the prognosis is usually miserable. Thus, our study aimed to investigate the prognostic factors in locally advanced NPC patients with ICA invasion.

## 2. Methods

### 2.1. Patients

This retrospective study collected and reviewed consecutive biopsy-proven NPC patients who received primary chemoradiotherapy from November 2015 to December 2022 at E-Da Hospital. All patients were scanned using either an MRI or CT and restaged in accordance with the 8th edition of the AJCC staging system for NPC [4]. This study was approved by the Institutional Review Boards at our hospital (EMRP18113N). The exclusion criteria included an incomplete scheduled CCRT course, an alive follow-up period of less than 12 months, and CT scans of the brain rather than MRI or CT scans of the head and neck before treatment. Common or internal carotid artery invasion only by a metastatic lymph node was also excluded. Fifty-four NPC patients with ICA invasion were initially included. To reduce the influence of distant metastases, 3 patients were excluded. A good treatment response was arbitrarily determined by a complete response/remission to the treatment or stable condition without progression either after primary treatment or after salvage treatment (chemoradiotherapy or surgical intervention). Four patients who died of second primary cancers within 36 months after primary NPC treatment and one patient who had a short follow-up of 12 months were excluded because the follow-up period was too short to determine the treatment response. Finally, a total of 47 cases were included in the group of ICA invasion for analyses. Patients were then stratified into a good treatment response and poor treatment response by performing receiver operation characteristic (ROC) curve analysis for continuous variables. The cut-off value of the NLR was also determined by ROC analysis and set at 2.9.

### 2.2. Image Assessment

Pre-treatment MRI and CT scans were reviewed by a radiologist experienced in head and neck images blinded to the clinical outcome. The diagnostic criteria for the common or internal carotid artery involvement were the following: (1) an encasement of 50% of the circumference of the common or internal carotid artery by the primary NPC (i.e., ≧180 degrees); (2) a segmental obliteration of the fat between the carotid artery and the primary NPC. In this study, both of the radiological features represented a sign of cancer involvement in the vascular wall and were classified as the carotid artery involvement group (Figure 1A–C). This 52-year-old male presented with left NPC (cT4N3aM0, IVB) and hematologic profiles including neutrophil 69.5%, lymphocyte 21.3%, and neutrophil-to-lymphocyte (NLR) 3.3. After primary chemoradiotherapy, the tumor invading the ICA had a complete remission and no recurrence at 41 months follow-up (Figure 1D–F).

### 2.3. Treatment Modality

All patients underwent radiotherapy based on IMRT with concurrent platinum chemotherapy according to our treatment guidelines. Patients received 2.00 Gy per fraction of radiotherapy, with a total of 70 Gy in 35 fractions. The planned dose of the gross tumor volume was 70 Gy, and for the lymph nodes it was 63 Gy. The radiotherapy was administered daily for 7 weeks. During CCRT, platinum was given concomitantly with radiotherapy. Patients received weekly cisplatin at 35 mg/m^2^ for a total of 7 weeks. For cisplatin-ineligible patients, carboplatin with AUC 2 was given weekly instead. Dose modification was performed according to their treatment toxicity. Computed tomography or magnetic resonance imaging was scheduled to evaluate the treatment response 1 month after CCRT, and every 2–3 months in the following days.

### 2.4. Statistical Analysis

The chi-square test was used to compare categorical variables, and Student’s *t*-test was used to compare continuous variables between groups. Receiver operating characteristic (ROC) curve analysis was performed for continuous variables to determine optimal cut-off points for distinguishing a good and poor treatment response. Survival probabilities were estimated by the Kaplan–Meier method, and the significance of differences was assessed by the log-rank test. To calculate the hazard ratio (HR) and the corresponding 95% confidence interval (CI) and to determine significant prognostic factors related to survival, multivariate analysis, based on univariate significance, was performed using the Cox proportional hazards regression model. *p* < 0.05 was considered statistically significant. Statistical analyses were conducted using SPSS Statistics version 24 (IBM, Armonk, NY, USA).

## 3. Results

### 3.1. Baseline Characteristics and Parameters

A total of 191 patients with NPC were included in this study. With a mean (±standard deviation) alive follow-up of 64.7 ± 22.5 months (range, 12–104 months), mortality events occurred in 53 (27.7%) patients. Among the 191 patients, 54 (28.3%) patients had ICA invasion by the primary nasopharyngeal tumor. Among patients with ICA invasion, 9 patients had T2 NPC and 45 patients had T3-T4 disease. T2 NPC extended into parapharyngeal space and invaded ICA. Hence, these 9 patients met our inclusion criteria and were included for analysis. The demographic and clinicopathological parameters and treatment outcomes were collected retrospectively, including age, sex, body mass index (BMI), neutrophil, lymphocyte, neutrophil-to-lymphocyte ratio (NLR) before the primary chemoradiotherapy, TNM classification (8th edition), treatment modalities, and death after treatment, and were compared between NPC patients with and without ICA invasion by NPC. After stratification, the results revealed that the T stage (*p* < 0.001), TNM stage (*p* = 0.002), and survival (*p* < 0.001) are significantly different between patients with and without ICA invasion groups in Table 1.

After performing ROC curve analysis for continuous variables, the optimal cut-off points for distinguishing a good and poor treatment response were determined in Appendix A. Patients were thereafter stratified according to the treatment response, with 25 patients in the poor response group and 22 patients in the good response group. The baseline characteristics and parameters related to survival were analyzed and a significant difference was found in the lymphocyte proportion (*p* = 0.008) and NLR (*p* = 0.008) between the good and poor response groups in Table 2.

### 3.2. Survival Outcomes

The ICA invasion group showed a worse prognosis compared to the no ICA invasion group (*p* < 0.001 in OS and DSS) in Figure 2. The 5-year OS was 43.4% and 84.9% in patients with and without ICA invasion, respectively (*p* < 0.001), in Table 1. The result in 5-year DSS was comparable with that in OS. When compared with patients without ICA invasion, the patients with ICA invasion by the NPC had a significantly worse 5-year OS and DSS (both *p* < 0.001; Figure 2). To understand the survival effect of the ICA invasion on the T stage, we further analyzed and found that better survival rates (OS and DSS) were observed in patients without ICA invasion in the T2 (*p* = 0.019 and *p* = 0.003), T3 (*p* = 0.006 and *p* = 0.003), and T4 (*p* = 0.003 and *p* = 0.005) stages compared to those with ICA invasion in Figure 3.

The outcomes of OS and DSS in the poor response group had a significant difference compared to the good response group (both *p* < 0.001) in Figure 4 and Table 2 (both *p* < 0.001). In univariate analysis, the lymphocyte (*p* = 0.040) or NLR (*p* = 0.040) was associated with OS and DSS (Table 3). In multivariate analysis, lymphocyte (*p* = 0.040) or NLR (*p* = 0.040) was an independently prognostic factor for OS (HR 2.430, 95% CI 1.040–5.678, *p* = 0.040 and HR 0.412, 95% CI 0.176–0.962, *p* = 0.040, respectively) and for DSS (HR 2.430, 95% CI 1.040–5.678, *p* = 0.040 and HR 0.412, 95% CI 0.176–0.962, *p* = 0.040, respectively). The data on OS are the same as DSS because three cases who died of lung cancer, prostate carcinoma, and stroke within 36 months and one case with a short follow-up have been excluded.

## 4. Discussion

To our best knowledge, our study is the first analysis to investigate the prognostic factors in locally advanced NPC patients with ICA invasion. In general, the ICA invasion group showed a worse prognosis compared to the no ICA invasion group (*p* < 0.001 in OS and DSS). Subgroup analysis showed that the inferior survival of locally advanced NPC with ICA invasion was observed across all T stages. In multivariate analysis, the NLR (*p* = 0.040) was the only independently prognostic factor for OS (HR 2.430, 95% CI 1.040–5.678, *p* = 0.040 and HR 0.412, 95% CI 0.176–0.962, *p* = 0.040, respectively) and for DSS (HR 2.430, 95% CI 1.040–5.678, *p* = 0.040 and HR 0.412, 95% CI 0.176–0.962, *p* = 0.040, respectively). Our study demonstrated that locally advanced NPC patients with ICA invasion had poor prognosis, especially those with a high NLR. Further prospective studies with larger cohorts are warranted to validate our results.

ICA invasion in patients with NPC is a significant adverse prognosticator [17]. However, the literature focused on ICA invasion is very limited, which suggests there is an unmet need in the treatment of NPC. The diagnosis of ICA invasion is usually according to imaging studies, including computed tomography or magnetic resonance imaging scans [18]. ICA invasion is generally associated with miserable outcomes, with reported OS rates ranging from 22% to 35% over 1 to 2 years depending on various clinical factors [11]. The management of ICA invasion is really challenging [14]. The surgical resection of NPC with carotid artery reconstruction is the main treatment [19]. A latest meta-analysis concluded that neck dissection with carotid artery reconstruction rather than ligation can improve survival and decrease the risk of carotid blow out syndrome [20]. For NPC patients with CAI, induction chemotherapy first followed by concurrent chemoradiotherapy succeed in prolonging survival as well as decreasing hemorrhage [14]. The most serious complication of ICA invasion is carotid blow out syndrome, which leads to massive bleeding and mortality [21]. Once carotid hemorrhage occurs, therapeutic embolization can save life with a lower risk of recurrent bleeding [21]. Seim et al. summarized several factors related to negative survival, including stage IV disease and an active tumor at carotid blow out, previously radiotherapy [22]. Given that several studies have confirmed the increasing risk of bleeding and unfavorable outcomes in NPC patients with CAI [14,15,16], there is an urgent need to explore the prognostic factors of NPC patients with ICA invasion. The reasons why NPC patients with CAI had poor prognosis were complicated. First, carotid artery blow out syndrome during chemoradiotherapy was the major reason, which led to the discontinuation of treatment or death. Second, some radiation oncologists reduced radiotherapy filed in the case of carotid blow out syndrome in NPC patients with CAI. This approach might compromise the local control rate, resulting in easy local recurrence in the future. Third, given that the tumor cell had invaded into the carotid artery, patients are prone to developing distant metastasis. All these causes contributed to the miserable prognosis in NPC patients with CAI.

Our study firstly identified that the NLR has a strong prognostication for locally advanced NPC patients with ICA invasion. Our conclusion provided clinical implications for physicians who treated patients with locally advanced NPC.

The NLR has been widely used as a potential prognostic biomarker for many solid tumors as well as in NPC [23]. Systemic review and meta-analysis confirmed that a high NLR is predictive of poorer survival in patients with NPC [24,25]. Furthermore, Lu et al. showed that the NLR in combination with the systemic inflammation index has significant clinical applications in the prognostic assessment of NPC [26]. Liu et al. proposed the NLR in combination with the monocyte-to-eosinophil ratio as a novel prognostic model for advanced nasopharyngeal carcinoma [27]. Kang et al. exhibited that the combined pretreatment NLR and platelet–lymphocyte ratio predict survival and prognosis in patients with non-metastatic nasopharyngeal carcinoma [28]. In the era of immunotherapy, Zhao et al. demonstrated that NLP has predictive value in patients with nasopharyngeal carcinoma receiving immune checkpoint inhibitors [29]. For NPC patients treated with CCRT, the NLR still provided a prognostic role to predict survival [30]. Meanwhile, NLP also could predict dysphagia severity and quality of life in NPC patients after radiotherapy [31]. A latest retrospective study comparing the prognostic values in severe inflammatory biomarkers among patients with NPC, which identified NLR as a most reliable and independent prognostic indicator for NPC patients receiving radiotherapy, offered valuable insights that could inform future clinical decision-making [32]. Our results are also consistent with the previous literature. The NLR could also be a strong prognosticator for locally advanced NPC patients with ICA invasion.

Our study has several inevitable biases which are inherent to any retrospective study. First, some patients developed carotid blow syndrome and discontinued their CCRT, resulting in negative outcomes. This will be a major bias in this study. Second, treatments between both groups were inconsistent. Patients received radiotherapy with 30–35Gy and chemotherapy with 35 mg/m^2^ weekly for 6–7 cycles. Finally, irregular follow-up intervals, inconsistent image modality, and different diagnostic criteria of ICA by radiologists would also limit the power of our study. Given several limitations, our study had clinical significance and provided real-world evidence for physicians who treated patients with NPC.

## 5. Conclusions

NPC patients with ICA invasion represent a significant negative prognostic factor that impacts treatment decisions and survival. Our study demonstrated that locally advanced NPC patients with ICA invasion had a significantly worse OS and DSS as compared with no ICA invasion patients. Subgroup analysis showed OS and DSS remained inferior in patients with ICA invasion to those in patients without ICA invasion regardless of the T stage. Patients were stratified into a poor response group and good response group. The NLR was significant between these groups. Multivariate analysis confirmed that the NLR was an independently prognostic factor for survival in locally advanced NPC patients with ICA invasion. By fostering a collaborative, multidisciplinary approach, healthcare providers can better navigate the challenges presented by ICA invasion in patients with NPC, aiming to enhance survival while minimizing complications and improving the quality of life for affected patients. Further prospective studies with a large cohort are warranted to validate our conclusions.

## Figures and Tables

**Figure 1 cancers-17-00488-f001:**
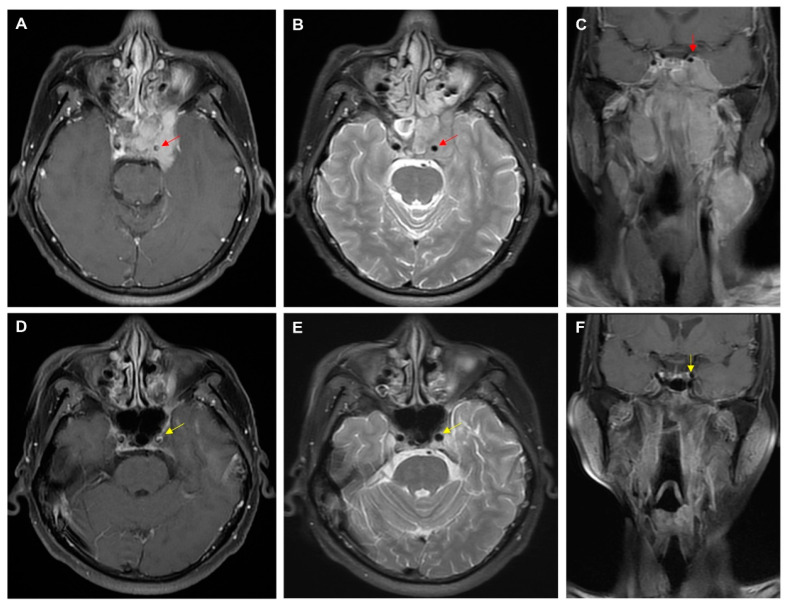
MRI scans showing internal carotid artery (ICA) invasion by nasopharyngeal carcinoma (NPC). Axial contrast-enhanced T1 weighted image (**A**), T2 weighted image (**B**), and coronal contrast-enhanced T1 weighted image (**C**), demonstrating at least 180 degrees of the ICA encasement with an obliteration of fat plane before treatment (red arrows). A complete response without recurrence 41 months after treatment shown on axial (**D**,**E**) and coronal images (**F**) (yellow arrows).

**Figure 2 cancers-17-00488-f002:**
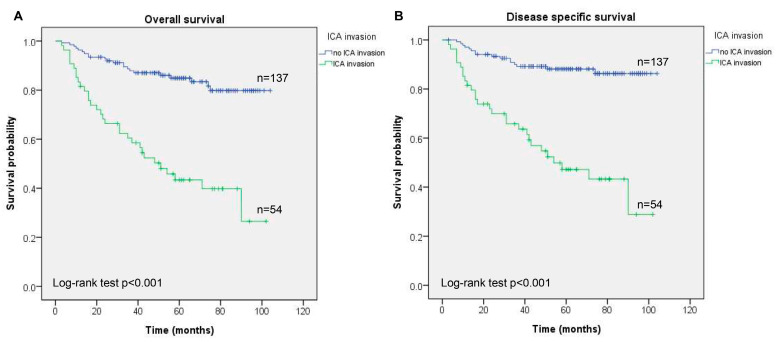
The Kaplan-Meier curves of (**A**) OS and (**B**) DSS for NPC patients with and without ICA.

**Figure 3 cancers-17-00488-f003:**
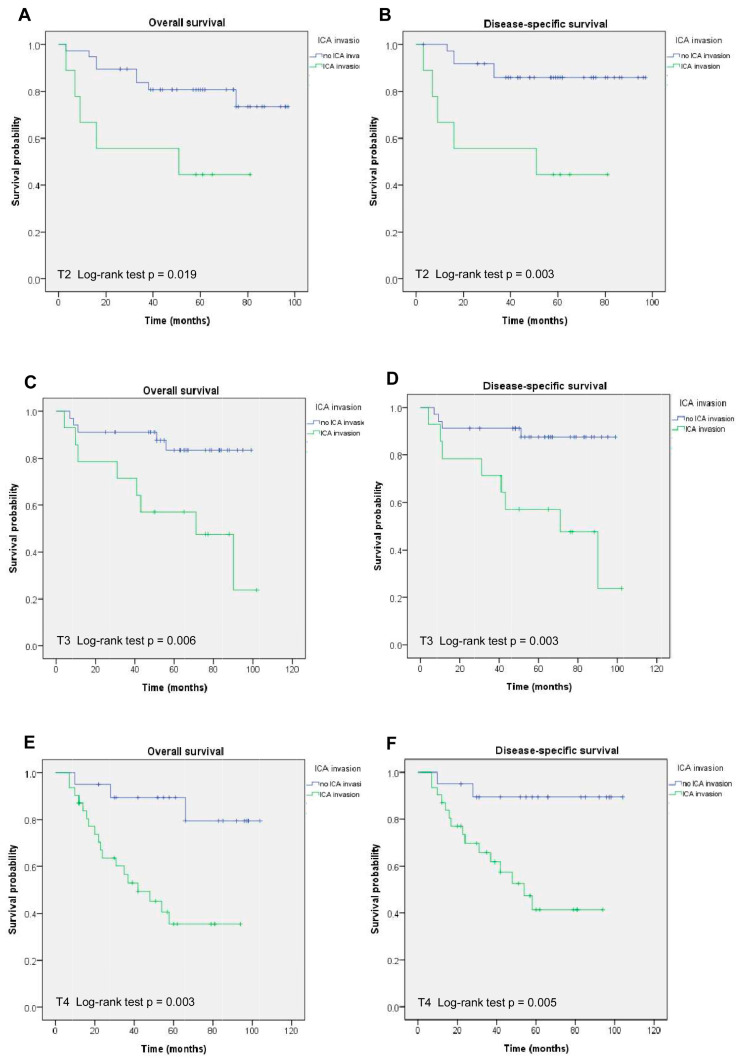
The Kaplan–Meier curves of OS and DSS stratified by T stage, including (**A**,**B**) T2, (**C**,**D**) T3, and (**E**,**F**) T4, for NPC patients with and without ICA invasion.

**Figure 4 cancers-17-00488-f004:**
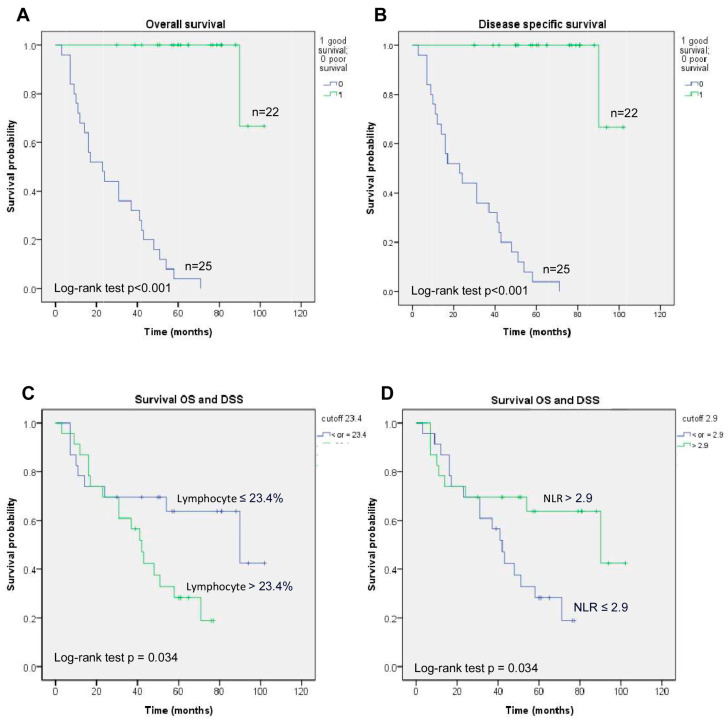
The Kaplan–Meier curves of (**A**) OS and (**B**) DSS for NPC patients with ICA invasion according to the treatment response and survival curves according to the (**C**) lymphocyte proportion and (**D**) NLR.

**Table 1 cancers-17-00488-t001:** Comparison of baseline characteristics and oncologic outcomes between nasopharyngeal carcinoma (NPC) patients with and without ICA invasion.

Variables	Patient	ICA Invasion	NO ICA Invasion	*p* Value
	N = 191	N = 54	N = 137	
Age, ≤55 years	109	26	83	0.118
>55 years	82	28	54	
Sex, Female	47	14	33	0.791
Male	144	40	104	
Smoke, No	88	27	61	0.494
Yes	103	27	76	
BMI, ≤25.3	109	33	76	0.478
>25.3	82	21	61	
T stage, T1T2	92	9	83	<0.001
T3T4	99	45	54	
N stage, N0N1	82	22	60	0.701
N2N3	109	32	77	
M stage, No	185	51	134	0.354
Yes	6	3	3	
TNM Stage, I/II	42	4	38	0.002
III/IV	149	50	99	
Neutrophil, ≤65.7	83	21	62	0.24
>65.7	96	32	64	
Lymphocyte, ≤23.4	79	28	51	129
>23.4	100	25	75	
NLR, ≤2.9	104	25	79	0.055
>2.9	75	28	47	
Alive	138	23	115	<0.001
Dead	53	31	22	
5-year OS	191	43.40%	84.90%	<0.001
Mean OS ± SD (months)		55.9 ± 5.4	90.5 ± 2.6	
5-year DSS	191	47.20%	88.20%	<0.001
Mean DSS ± SD (months)		58.9 ± 5.5	93.9 ± 2.4	

BMI, body mass index; DSS, disease-specific survival; ICA, internal carotid artery; NPC, nasopharyngeal carcinoma; NLR, neutrophil-to-lymphocyte ratio; OS, overall survival; RT, radiotherapy.

**Table 2 cancers-17-00488-t002:** Comparison of baseline characteristics and oncologic outcomes between patients with good treatment response and with poor treatment response.

Variables	Patient	Poor Treatment Response (25)	Good Treatment Response (22)	*p* Value
	No. (47)			
Age, ≤55 years	23	9	14	0.059
>55 years	24	16	8	
Sex, Female	12	7	5	0.679
Male	35	18	17	
Smoke, No	23	9	14	0.059
Yes	24	16	8	
BMI, ≤25.3	28	13	15	0.259
>25.3	19	12	7	
T stage, T2	9	5	4	0.874
T3T4	38	20	18	
N stage, N0N1	19	13	6	0.085
N2N3	28	12	16	
Stage, I/II	4	2	2	0.894
III/IV	43	23	20	
Neutrophil, ≤65.7	20	14	6	0.062
>65.7	26	11	15	
Lymphocyte, ≤23.4	23	8	15	0.008
>23.4	23	17	6	
NLR, ≤2.9	23	17	6	0.008
>2.9	23	8	15	
5-year OS	47	4.00%	100.00%	<0.001
Mean OS ± SD (months)		27.3 ± 3.8	98.0 ± 3.3	
5-year DSS	47	4.00%	100.00%	<0.001
Mean DSS ± SD (months)		27.3 ± 3.8	98.0 ± 3.3	

BMI, body mass index; DSS, disease-specific survival; ICA, internal carotid artery; NPC, nasopharyngeal carcinoma; NLR, neutrophil-to-lymphocyte ratio; OS, overall survival; RT, radiotherapy.

**Table 3 cancers-17-00488-t003:** Univariate and multivariate analyses of prognostic factors in 47 NPC patients with ICA invasion.

Variables	OS		DSS	
	Univariate	Multivariate	Univariate	Multivariate
	HR (95% CI)	HR (95% CI)	HR (95% CI)	HR (95% CI)
Age, ≤55 years	*p* = 0.058		*p* = 0.058	
>55 years	0.464 (0.209–1.027)		0.464 (0.209–1.027)	
Sex, Female	*p* = 0.852		*p* = 0.852	
Male	0.920 (0.384–2.206)		0.920 (0.384–2.206)	
Smoke, No	*p* = 0.113		*p* = 0.113	
Yes	0.519 (0.231–1.168)		0.519 (0.231–1.168)	
BMI, ≤25.3	*p* = 0.468		*p* = 0.468	
>25.3	1.331 (0.615–2.879)		1.331 (0.615–2.879)	
T stage, T2	*p* = 0.667		*p* = 0.667	
T3T4	0.806 (0.301–2.155)		0.806 (0.301–2.155)	
N stage, N0N1	*p* = 0.126		*p* = 0.126	
N2N3	0.545 (0.251–1.186)		0.545 (0.251–1.186)	
Stage, I/II	*p* = 0.785		*p* = 0.785	
III/IV	1.223 (0.287–5.207)		1.223 (0.287–5.207)	
Neutrophil, ≤65.7	*p* = 0.164		*p* = 0.164	
>65.7	0.568 (0.256–1.259)		0.568 (0.256–1.259)	
Lymphocyte, ≤23.4	*p* = 0.040	*p* = 0.040	*p* = 0.040	*p* = 0.040
>23.4	2.430 (1.040–5.678)	2.430 (1.040–5.678)	2.430 (1.040–5.678)	2.430 (1.040–5.678)
NLR, ≤2.9	*p* = 0.040	*p* = 0.040	*p* = 0.040	*p* = 0.040
>2.9	0.412 (0.176–0.962)	0.412 (0.176–0.962)	0.412 (0.176–0.962)	0.412 (0.176–0.962)

CI, confidence interval; HR, hazard ratio; BMI, body mass index; DSS, disease-specific survival; ICA, internal carotid artery; NPC, nasopharyngeal carcinoma; NLR, neutrophil-to-lymphocyte ratio; OS, overall survival.

## Data Availability

These data are unavailable due to privacy or ethical restrictions.

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
