# Peer review of "Analysis of Prognostic Factors for Internal Carotid Artery Invasion by Nasopharyngeal Carcinoma"

_cancers, 2025, doi:10.3390/cancers17030488_

Round 1
Reviewer 1 Report
Comments and Suggestions for Authors
I read with great interest this manuscript which deals with the problems of nasopharyngeal carcinomas with carotid involvement. I compliment the authors for the well-presented series and the impeccable conclusions.
I only have two observations to make.
In the introduction, lines 42-44, the authors write: ““The highest rates of NPC are found in Southeast Asia, particularly in southern China, Malaysia, and Singapore”.
In presenting the epidemiological characteristics of NPC, I believe it is necessary to add the different incidence of NK-NPC and K-NPC in Southeast Asia compared to Western countries.
Second, looking at Table 1, I wonder how it was possible that 9 patients in stage T1-2 had carotid invasion, since patients with common carotid invasion only by N were excluded (lines 81-82 “Common or internal carotid artery invasion only by a metastatic lymph node was also excluded”). If I understand the 8th TNM classification correctly, a NPC with invasion of the intracranial carotid artery from the primary tumor is, by definition, a T4. Where was the carotid involved in these T1-2?
Author Response
Dear editors and reviewers:
Thank you for your kind review. Here are our responses to reviewer #1.
In the introduction, lines 42-44, the authors write: ““The highest rates of NPC are found in Southeast Asia, particularly in southern China, Malaysia, and Singapore”. In presenting the epidemiological characteristics of NPC, I believe it is necessary to add the different incidence of NK-NPC and K-NPC in Southeast Asia compared to Western countries.
Response: Thank you for your suggestion. We had added the incidence of keratinizing and non-keratinizing NPC between Asia and Western countries. As mentioned in Line 54-58, Meanwhile, the incidences of keratinizing NPC and non-keratinizing NPC were different between Southeast Asia and Western countries. In endemic area, 25% patients were ke-ratinizing NPC and 75% were non-keratinizing NPC [4], while in non-endemic area, 45% were differentiated NPC (including keratinizing and non-keratinizing) and 55% were undifferentiated NPC [5].
Second, looking at Table 1, I wonder how it was possible that 9 patients in stage T1-2 had carotid invasion, since patients with common carotid invasion only by N were excluded (lines 81-82 “Common or internal carotid artery invasion only by a metastatic lymph node was also excluded”). If I understand the 8th TNM classification correctly, a NPC with invasion of the intracranial carotid artery from the primary tumor is, by definition, a T4. Where was the carotid involved in these T1-2?
Response: Thank you for your question. According to AJCC 8th edition, T4 refers to the main tumor has grown inside the skull and/or into the cranial nerves, the hypopharynx, the main salivary gland, or the eye or its nearby tissues. Accurately, there is no standard definition of T stage regarding NPC involving carotid artery. Among our patients, some T1-T2 NPC tumor invaded extra-cranial internal carotid artery which met our inclusion criteria. This data was presented in Table 1.
Reviewer 2 Report
Comments and Suggestions for Authors
This is a very excellent paper. Your own data show that internal carotid artery invasion in nasopharyngeal carcinoma has a poor prognosis.
I have a question.
It is predictable that internal carotid artery invasion has a poor prognosis, and the results are easy to accept, but is this due to poor local control or is it due to arterial invasion, distant metastasis, etc.?
If you have the data, we would appreciate it if you could show us the results.
Following that, I would like to see a little more discussion on why carotid artery invasion has a poor prognosis. I am sure there are many points to discuss clinically, pathologically, molecularly, and so on.
In addition, I would like you to describe how to improve the prognosis of such cases of internal carotid artery invasion.
What about advances in radiotherapy equipment as modality, for example, proton beams and heavy ion radiotherapy?
In addition, if you could include more information on the efficacy of molecular targeted therapies, immune checkpoint inhibitors, etc., this paper would be even more impressive and worthy of inclusion in the Cancers. Thank you in advance.
Author Response
Dear editors and reviewers:
Thank you for your kind review. Here are our responses to reviewer #2.
It is predictable that internal carotid artery invasion has a poor prognosis, and the results are easy to accept, but is this due to poor local control or is it due to arterial invasion, distant metastasis, etc.? If you have the data, we would appreciate it if you could show us the results. Following that, I would like to see a little more discussion on why carotid artery invasion has a poor prognosis. I am sure there are many points to discuss clinically, pathologically, molecularly, and so on.
Response: Thank you for your questions. We had added some discussion regarding to the poor prognosis of NPC patients with carotid artery invasion. As added in Line 239-246, the reasons why NPC patients with CAI were complicated. First, carotid artery blow out syndrome during chemoradiotherapy was the major reason, which led to discontinue treatment or death. Second, some radiation oncologists reduced radiotherapy filed in case of carotid blow out syndrome in NPC patients with CAI. This approach might compromise the local control rate, resulting in easily local recurrence in the future. Third, given that tumor cell had invaded into carotid artery, patients prone to develop distant metastasis All these causes contributed the miserable prognosis in NPC patients with CAI.
In addition, I would like you to describe how to improve the prognosis of such cases of internal carotid artery invasion.
Response: Thank you for your suggestion. We had added some discussion about how to improve and management of carotid artery invasion in our manuscript. As mentioned in line 226-234, the management of ICA invasion is really challenging [14]. Surgical resection of NPC with carotid artery reconstruction is the main treatment [19]. A latest meta-analysis concluded that neck dissection with carotid artery reconstruction rather than ligation can improve survival and decrease the risk of carotid blow out syndrome[20]. For NPC patients with CAI, induction chemotherapy first followed by concurrent chemoradiotherapy succeed in prolonging survival as well as less hemorrhage [14]. The most serious complication of ICA invasion is carotid blow out syndrome, which led to massive bleeding and mortality [21]. Once carotid hemorrhage, therapeutic embolization can save life with lower risk of recurrent bleeding [21].
What about advances in radiotherapy equipment as modality, for example, proton beams and heavy ion radiotherapy?
Response: Thank you for your question. However, little was known regarding whether novel radiotherapy equipment could benefit NPC patients with carotid artery invasion or decrease risk of bleeding. Hence, we can't provide any comments about this issue.
In addition, if you could include more information on the efficacy of molecular targeted therapies, immune checkpoint inhibitors, etc., this paper would be even more impressive and worthy of inclusion in the Cancers. Thank you in advance.
Response: Thank you for your suggestion. Currently, more and more immune checkpoint inhibitors provide survival benefits in NPC patients. However, our manuscript aimed to explore the prognostic impact of carotid artery invasion in NPC patients. There were no clear evidence regarding the role of novel agents, including molecular targeted therapies and immune checkpoint inhibitors, on NPC patients with carotid artery invasion. Hence, we can't have more discussion on this issue.
Round 2
Reviewer 2 Report
Comments and Suggestions for Authors
This is important paper for our HN surgical oncologist. Thank you so much.
Author Response
This is important paper for our HN surgical oncologist. Thank you so much.
Response: Thank you for your kind review.